# Synergistic Immunosuppression of Avian Leukosis Virus Subgroup J and Infectious Bursal Disease Virus Is Responsible for Enhanced Pathogenicity

**DOI:** 10.3390/v14102312

**Published:** 2022-10-21

**Authors:** Weiguo Chen, Sheng Chen, Yu Nie, Wenxue Li, Hongxin Li, Xinheng Zhang, Feng Chen, Qingmei Xie

**Affiliations:** 1Heyuan Branch, Guangdong Provincial Laboratory of Lingnan Modern Agricultural Science and Technology & Guangdong Provincial Key Laboratory of Agro-Animal Genomics and Molecular Breeding, College of Animal Science, South China Agricultural University, Guangzhou 510642, China; 2Guangdong Engineering Research Center for Vector Vaccine of Animal Virus, Guangzhou 510642, China; 3South China Collaborative Innovation Center for Poultry Disease Control and Product Safety, Guangzhou 510642, China; 4Key Laboratory of Animal Health Aquaculture and Environmental Control, Guangzhou 510642, China

**Keywords:** Avian Leukosis Virus Subgroup J (ALV-J), Infectious Bursal Disease Virus (IBDV), superinfection, pathogenicity, immunosuppression, lymphocyte subsets

## Abstract

In recent years, superinfections of avian leukosis virus subgroup J (ALV-J) and infectious bursal disease virus (IBDV) have been frequently observed in nature, which has led to the increasing virulence in infected chickens. However, the reason for the enhanced pathogenicity has remained unclear. In this study, we demonstrated an effective candidate model for studying the outcome of superinfections with ALV-J and IBDV in cells and specific-pathogen-free (SPF) chicks. Through in vitro experiments, we found that ALV-J and IBDV can establish the superinfection models and synergistically promote the expression of IL-6, IL-10, IFN-α, and IFN-γ in DF-1 and CEF cells. In vivo, the weight loss, survival rate, and histopathological observations showed that more severe pathogenicity was present in the superinfected chickens. In addition, we found that superinfections of ALV-J and IBDV synergistically increased the viral replication of the two viruses and inflammatory mediator secretions in vitro and in vivo. Moreover, by measuring the immune organ indexes and blood proportions of CD3^+^, CD4^+^, and CD8α^+^ cells, our results showed that the more severe instances of immunosuppression were observed in the superinfected chickens. In the present study, we concluded that the more severe immunosuppression induced by the synergistic viral replication of ALV-J and IBDV is responsible for the enhanced pathogenicity.

## 1. Introduction

Complex infections, which occur when at least two different pathogens establish an infection, have broad implications for the evolution of virulence, genetic diversity, epidemiology, and control strategies. Coinfections and superinfections refer to how complex infections develop before or after the development of an adaptive immune response [1,2]. In some studies, however, this semantic point causes confusion and sometimes complicates comparisons between studies [3]. The outcome of any coinfection or superinfection is complicated and affected by the interactions between the infectious agents, the characteristics of the host, the microorganism community, and the triggered immune response [4]. Thus, different scenarios can be observed, including the first pathogen promoting a more virulent infection of the second, reducing its severity, or completely suppressing it [5]. Conversely, the second pathogen may also directly or indirectly influence the first. It is this complexity that makes the common reductionist approach of host–pathogen interactions unsuitable for the study of single infections [6]. On a special note, when the first pathogen can cause immunosuppression in the host, the second one usually causes stronger progress of the disease. For instance, people with human immunodeficiency virus (HIV) superinfections have a lower CD4^+^ cell count, which leads to faster disease progression [7,8]. There are, however, a number of fundamental knowledge gaps to be filled, including how virulence changes in hosts, especially in superinfection.

Superinfection is not only an important infection model in medicine but also a serious threat to the development of the poultry industry. Several studies have demonstrated that chickens are often infected by several pathogens simultaneously, which usually cause more severe clinical symptoms and mortality [2,9,10,11]. For example, chicks dually infected with chicken anemia virus (CAV) and reovirus demonstrated the synergism of lower weight gain and more severe tissue damage [12]. Marek’s disease virus (MDV) and reticuloendotheliosis virus (REV) superinfections significantly increased disease severity and reduced vaccine efficacy [11]. These results indicate that the effect of a superinfection is greater than the mere sum of its parts. Among them, ALV-J in particular is more likely to induce complex infections due to its viral characteristics.

ALV-J can cause neoplastic disease and immunosuppression in infected chickens, resulting in huge economic losses for the poultry industry worldwide. A remarkable property of ALV-J is its vertical transmission, which is responsible for persistent and complex infections [13,14]. For this reason, one-day-old chickens are often infected with ALV-J, and it makes these flocks more susceptible to secondary viruses, a process known as superinfection, which causes immunosuppression, growth retardation, and tissue tumors in the infected chickens [15,16]. Several reports have reported that ALV-J superinfections with REV or MDV caused more severe growth arrest and immunosuppression than single infections [2,10]. IBDV, like ALV-J, leads to severe immunosuppression in young chickens. At 3–6 weeks of age, chickens are most susceptible to IBDV due to this period’s coincidence with the stage of Bursa Fabricii development. Therefore, ALV-J and IBDV are often cause superinfections in poultry production [17]. However, the cellular and animal models of ALV-J and IBDV superinfections have not been established. In addition, the changes in virulence and the degree of immunosuppression of superinfection are still unclear.

In this research, we present a systematic and in-depth analysis of the pathogenicity and immunosuppression activity of superinfections with ALV-J and IBDV in vitro and vivo. The purpose of the present work was to establish superinfection models and examine the effects of superinfections with ALV-J and IBDV.

## 2. Materials and Methods

### 2.1. Viruses, Cells, and Animals

ALV-J strain SCAU-HN06 was a generous gift from Professor Liao of South China Agricultural University. IBDV strain 801 (cell-adapted strain) was stored in our laboratory. Virus titers were calculated using Reed–Muench formula to calculate 50% tissue-cultural-infective dose (TCID_50_)/mL. DF-1 cells were stored in our laboratory and cultured in Dulbecco’s Modified Eagle’s Medium (DMEM, Thermo Fisher Scientific, Waltham, MA, USA), supplemented with 10% fetal bovine serum (FBS, Gibco, Waltham, MA, USA), and maintained at 37 °C and 5% CO_2_. Chicken embryo fibroblasts (CEF) cells were prepared as previously described [18].

### 2.2. Cell Experimental Design

DF-1 and CEF cells were divided into 4 groups in 6-well dishes, which included experimental treatment groups ALV-J, IBDV, ALV-J+IBDV, and control. In ALV-J and ALV-J+IBDV groups, cells were infected with ALV-J at a multiplicity of infection (MOI) of 1. After 24 h post-ALV-J infection, cells were infected with IBDV at MOI of 0.1. The cells from control group were treated with equal amount PBS. Samples were collected at 12, 24, 36, and 48 h post-IBDV infection. To determine the viral proliferation, total cell samples were collected for RT-qPCR and protein samples for Western Blot detection. Cell supernatants were collected to measure the concentrations of cytokines by ELISA. Specific groups and treatments are shown in Figure 1A.

### 2.3. ELISA for Cytokines

A 5 mL supernatant was collected from samples at 12, 24, 36, and 48 hpi and then centrifuged at 1000 rpm for 20 min. To measure the expression levels of IL-6, IL-10, IFN-α, and IFN-γ in the sample, ELISA tests were carried out using commercial kits (CUSABIO, Houston, TX, USA) according to the manufacturer’s instructions.

### 2.4. Quantitative Reverse Transcription-PCR

Quantitative reverse transcription-PCR (RT-qPCR) was used to detect viral loads and expression levels of cytokine genes. The total RNA of samples was extracted by Trizol Reagent (Invitrogen, Waltham, MA, USA), while qPCR experiments were performed using Taq Pro Universal SYBR qPCR Master Mix (Vazyme, Nanjing, China) according to the manufacture’s instruction. Results were analyzed using the 2^−ΔΔCt^ method. The primer sequences used for amplifications are listed in Table 1.

### 2.5. Immunofluorescence and Confocal Microscopy

DF-1 and CEF cell monolayers were fixed using 4% paraformaldehyde (Python Bio, Guangzhou, China) for 15 min at room temperature. Then, cells were permeabilized with 0.1% Triton-X 100 (Python Bio, Guangzhou, China) for 15 min at room temperature. Afterwards, cells were blocked with 5% skimmed milk powder for 30 min at room temperature. Next, cells were incubated with ALV-J Env antibody (JE9) (generous gift from Professor Qin of Yangzhou University) and IBDV-VP2 antibody (stored in our lab) for 1 h and then fluorochrome-conjugated secondary antibodies (Alexa fluor 488 for ALV-J Env and Alexa fluor 594 for IBDV VP2) (Abcam, Cambridge, UK) for 1 h in the dark. For nuclear staining, cells were incubated with DAPI (Beyotime, Shanghai, China). The fluorescence signals were visualized with a TCS SP8 confocal fluorescence microscope (Leica Microsystems GmbH, Wetzlar, Germany).

### 2.6. Animal Experimental Design

A total of 220 one-day-old SPF chicks were randomly divided into 4 groups, which included experimental treatment groups ALV-J, IBDV, and ALV-J + IBDV, and control. At 1 day of age, chicks in the ALV-J and IBDV+ALV-J groups were inoculated via intra-abdominal injection with ALV-J at 100 μL 10^3.7^ TCID_50_. At 14 days of age, chicks in the IBDV and IBDV + ALV-J groups were inoculated intra-abdominally with IBDV at 100 μL 10^3.6^ ELD_50_. The chicks from control group were inoculated with 100 μL PBS at 1 day of age and 14 days of age. Weight loss and mortality were recorded throughout the experimental period. At 3, 7, 14, and 21 days post-infection (dpi), venous blood samples were collected from three chickens of each group and virus titers were detected by RT-qPCR. Considering the accidental death caused by inoculation in abdomen, chicks dead in the first 3 days were subtracted from total number of birds. The animal experiment was repeated to test the consistency and the reliability between two independent experiments. Specific groups and treatments are shown in Figure 1B.

### 2.7. Index of Immune Organ

At 7 and 21 dpi, samples of the immune organs including Bursa Fabricii and spleen from 6 chickens in each group were excised and weighed. A measurement of the immune organs’ indecies can be obtained by comparing the weight of the immune organ to the weight of the body.

### 2.8. Histopathological and Examination

At 21 dpi, the immune organs (including spleen, Bursa Fabricii, and thymus) and other organs (including heart, liver, and kidneys) were stained by standard hematoxylin-eosin procedure for histological observation under light microscope. After collecting the tissues and fixing them in 10% formalin for 120 h, paraffin sections with a thickness of 5 mm were cut. Hematoxylin and eosin staining was performed after deparaffinization and hydration of slides.

### 2.9. Fluorescent Cell Sorting Analysis of Peripheral Blood Lymphocyte Subpopulations

At 21 dpi, lymphocytes from three chickens from each group were isolated using lymphocyte separation kit (TBD science, Tianjin, China) according to the manufacturer’s protocol. A total of 10^6^ lymphocytes were incubated with 3 μL of PE-CD3 antibody (1:100) (MA5-28697, Invitrogen, USA), 2 μL of FITC-CD4 antibody (1:100) (MA5-28685, Invitrogen, USA), and 1 μL of APC-CD8 antibody (1:100) (MA5-28718, Invitrogen, USA). After mixing, the samples were incubated in the dark for 30 min at room temperature. A total of 1 mL of PBS was added to each tube, and the cells were collected by centrifugation at 400× *g* for 5 min. Then, the supernatant was discarded, and the cell pellet was washed twice with PBS. A total of 300 µL of PBS was added to each tube for flow analysis. The data were analyzed by flow cytometry using a fluorescence-activated-cell-sorting (FACS) BD AccuriC6 cell sorter (Becton, Dickinson and Company, New Jersey, NJ, USA).

### 2.10. Statistical Analysis

The statistical analysis was conducted using GraphPad Prism version 8.0 (San Diego, CA, USA). Statistically significant differences between multiple experiment groups were determined using one-way analysis of variance (ANOVA) and Tukey’s test. Different lowercase letters indicate significant differences between different groups. Comparisons of the viral-titer and viral-load data between two groups were performed using Student’s *t*-test. Survival curves between two groups were compared using a log-rank test (Mantel–Cox). Differences were considered statistically significant at *p* < 0.05.

## 3. Results

### 3.1. ALV-J and IBDV Synergistically Increase Viral Replication In Vitro

The DF-1 cell line is widely accepted as a common cell line used for avian virus studies and both ALV-J and IBDV can infect DF-1 alone. CEF, as primary culture cells, can better reflect the interaction between the pathogen and host. To investigate the synergistic effects of ALV-J and IBDV, monolayer CEF and DF-1 cells were inoculated with PBS, ALV-J, IBDV, and both viruses (ALV-J + IBDV). The time course of the superinfection is shown in Figure 1A. Firstly, before measuring the viral replication, we used laser-confocal microscopy to determine whether the two viruses could cause a superinfection. As shown in Figure 2A, ALV-J and IBDV showed obvious colocalization in both DF-1 and CEF cells, indicating the successful formation of a superinfection. Since the level of infection varies, the accumulation level of ALV-J and IBDV was significantly higher in the superinfection group than that in the single-virus-infected group at 12 hpi to 36 hpi. However, after 48 hpi, the viral titers in the IBDV-infected group and the superinfection group suddenly decreased to a very low level due to accelerated cell death (Figure 2B–E). We next evaluated the protein level of the two viruses via Western blotting at 36 hpi (Figure 2F,G). The results showed that both the ALV-J and IBDV protein expressions were significantly increased in the superinfected group at 36 hpi, whether in DF-1 or CEF cells (Figure 2I,J). In addition, the immunofluorescence assay was also consistent with the above results (Figure 2H). Taken together, the superinfection of ALV-J and IBDV synergistically increased viral replication in vitro.

### 3.2. ALV-J and IBDV Synergistically Induce Inflammatory Mediator Secretion In Vitro

The balance of inflammatory factors is also an important index to evaluate the level of cellular immunity. To further understand whether a superinfection of ALV-J and IBDV could synergistically induce inflammatory mediator secretion in vitro, we tested the dynamic changes of several inflammatory mediators using ELISA. The results showed that the superinfected cells were found to have significantly higher levels of IL-6, IL-10, IFN-α, and IFN-γ compared to the single-virus-infected group or the controls (Figure 3). These results showed that the superinfection of ALV-J and IBDV indeed induces inflammatory mediator secretion in vitro.

### 3.3. ALV-J and IBDV Synergistically Enhance Pathogenicity in SPF Chickens

To further determine the synergistic effects of a superinfection of ALV-J and IBDV in vivo, we performed animal experiments on the SPF chickens as documented in Figure 1B. No chickens showed clinical symptoms and mortality in the control group. The survival period of the chickens in the superinfection group of ALV-J and IBDV was generally shorter than that of the chickens in the ALV-J or IBDV singly infected groups, as shown by the survival curves in Figure 4A. The overall mortality rate of the ALV-J- and IBDV-infected chickens was 16% and 44%, respectively, while that of the superinfection group was 72%. By continuously measuring the weight gain of chickens for 49 days, we found that chickens in the superinfection group gained weight more slowly, weighing only 253.57 ± 15.9 g at 49 days, much less than the 463.48 ± 22.56 g of the control group. The body weights of the ALV-J- or IBDV-infected groups at 49 days were 328.29 ± 26.11 g and 301.12 ± 15.59 g, respectively (Figure 4B). Our histopathological results have also confirmed an enhanced pathogenicity. In the superinfection group, a more severe inflammatory cell infiltration was observed in the heart, liver, kidneys, and Bursa Fabricii; additionally, the heterophilic granulocyte levels had increased and connective tissue hyperplasia was observed (Figure 4C). These results showed that the severity of the pathogenicity, including weight loss and mortality, caused by the superinfection was more serious than that caused by ALV-J or IBDV alone.

### 3.4. ALV-J and IBDV Synergistically Increase Viral Replication In Vivo

Although we have a clear understanding of superinfection synergistically increasing viral replication in vitro, the nature of viral proliferation in vivo is unclear. To further analyze the viral titers in the tissues and blood of infected chickens, the samples were tested at 3, 7, 14, and 21 dpi by RT-qPCR. The results showed that the viral load of ALV-J or IBDV in the blood of the superinfection group was significantly increased far more than in the single-virus-infected chickens from 3 dpi to 21 dpi (Figure 5A,B). Similarly, in different tissues, the viral loads of both ALV-J and IBDV in the superinfection group were also far more than the single virus-infected group (Figure 5C,D). Most interestingly, the synergistic viral replication of ALV-J and IBDV in the superinfected chickens is much higher than those in the CEF and DF-1 cells. Based on these results, ALV-J infection is capable of affecting IBDV replication in vivo.

### 3.5. ALV-J and IBDV Synergistically Induce Immunosuppression in SPF Chickens

Both ALV-J and IBDV are immunosuppressive viruses, and an infection of either one can cause severe immunosuppression in chickens. To determine whether a superinfection of ALV-J and IBDV can synergistically induce more severe immunosuppression in chickens, we evaluated the index of the immune organs, the differentiation of lymphocyte subsets, and the secretion of inflammatory factors. The anatomical results showed that the superinfection of ALV-J and IBDV induced the atrophy of immune organs (Figure 6A). The immune organ index results showed that the index of the spleen in the superinfection group exhibited no significant change at 7 dpi, while the index of Bursa Fabricii decreased significantly (Figure 6B). However, at 21 dpi, the indexes of the spleen and Bursa Fabricii in the superinfection group were much lower than those in the single virus-infected group (Figure 6C).

Next, we investigated the cellular immune responses by measuring the proportions of CD3^+^, CD4^+^, and CD8α^+^ cells in the blood (Figure 7A). Our results showed that the CD3^+^, CD4^+^, and CD8α^+^ cell proportions slightly decreased in the ALV-J single virus-infected group, but not significantly, while those in the IBDV single virus-infected group showed a significant decrease compared to the control group. In the superinfection group, the lymphocyte cells exhibited a more drastic decline than that of the control group, which was not significant compared to that of the IBDV single virus-infected group. However, when compared with the controls, the CD4^+^/CD8^+^ ratios were similar in all the infected groups. In addition, by detecting the expression levelss of IL-6, IL-10, IFN-α, and IFN-γ in several immune organs, including the spleen, thymus, and Bursa Fabricii, we found that the inflammatory factors were significantly increased in all the infected groups. Moreover, the expression level of inflammatory factors in the superinfection group showed a significant increase compared to that of the single virus-infected group, indicating a more severe degree of immunosuppression induced by the superinfection. In addition, the results in vivo were also consistent with those in vitro. These results showed that the superinfection of ALV-J and IBDV induces more severe immunosuppression than that in a single virus-infected group in chickens.

## 4. Discussion

In recent years, although the eradication of ALV-J has been continuously carried out, the outlook regarding the prevention and control of ALV-J is still pessimistic due to many problems such as complex genetic resources, the serious contamination of commercial vaccines, and poor breeding environments [18,19]. An important feature of an ALV-J infection is that it causes severe immunosuppression in infected chickens [20]. Moreover, due to ALV-J being easily transmitted vertically, it is likely to induce superinfections. Therefore, it is currently accepted that a superinfection of ALV-J ultimately leads to increased virulence and reduced vaccine efficacy in infected chickens [21,22]. As mentioned above, ALV-J is often mixed with homologous viruses such as ALV-A, ALV-B, and ALV-K as well as heterologous viruses such as REV [23], MDV [2], CAV [22], and IBDV. There are many natural examples of ALV-J superinfections with different viruses; however, the direct cause for the enhanced pathogenicity and tumorigenesis remains obscured.

In the present study, we demonstrated a synergism in the viral replication of a heterologous superinfection of ALV-J and IBDV in vitro and in vivo. Notably, our results showed that chickens superinfected with ALV-J and IBDV not only caused much more severe growth retardation and mortality, but also more severe immunosuppression (Figure 6 and Figure 7). In viral synergistic interactions, a specific crosstalk among different viral proteins modulates the natural history, the immune response, and life cycle of both viruses in mixed-infected individuals [24,25]. These effects are mediated by immune mechanisms and crosstalk between the two viruses, which can interfere with host defense mechanisms [2,26]. In particular, when one or both viruses are immunosuppressive, more severe immunological defects are observed. For instance, several studies have shown that HIV worsens the course of a hepatitis C virus (HCV) infection, increasing the risk of disease. In addition, HCV may increase immunological defects due to HIV [26].

To verify the hypothesis that the more severe immunosuppression induced by the synergistic viral replication of ALV-J and IBDV is the direct cause of the enhanced pathogenicity, we tested the host immune responses in vivo in ALV-J and IBDV dually infected chickens. First, immune organ indexes are the most commonly used measures of immunity in poultry. In the present study, we found that the Bursa Fabricii of the group superinfected with ALV-J and IBDV exhibited significantly greater atrophy and dehydration than the control group or any single virus-infected group. The original feature of IBDV infection is bursal necrosis, which causes severe damage to the immune system [27]. These results suggest that the superinfection of IBDV induced by ALV-J may aggravate the damage to the Bursa Fabricii and cause more severe immunosuppression. A previous study has shown that bursal lesions are associated with the degree of B cell depletion [28,29]. Moreover, ALV-J infection could cause the developmental arrest and dysfunction of B cell progenitors in the Bursa Fabricii [30], which may be synergistic with IBDV-induced bursal injury, resulting in more severe immunosuppression in superinfected chickens. On the other hand, it is well known that ALV-J infection causes splenomegaly in chickens [31], while IBDV infection causes spleen atrophy [32]. In the superinfection group, the spleen showed significantly greater atrophy compared to the ALV-J infected group at 21 dpi, which illustrates the role of IBDV in the superinfection group. Thus, both ALV-J and IBDV can induce immune organ injury in SPF chickens, while additive effects were also observed.

With respect to inflammatory mediator secretion, both ALV-J and IBDV dramatically upregulated IL-6, IL-10, IFN-α, and IFN-γ expressions in the spleen, Bursa Fabricii, and thymus. The balance of inflammatory factors is also an important index to evaluate the level of cellular immunity. The superinfection of ALV-J and IBDV promotes the secretion of inflammatory factors in vitro and vivo, which is also a manifestation of the synergistically enhanced pathogenicity of both viruses. These results were consistent with those of previous studies [9,33,34], and our histopathology results confirmed this. We observed necrosis and the depletion of lymphoid cells of the Bursa Fabricii, inflammatory cell infiltration, and fibroplasias in the superinfection group and single virus-infected groups, which was consistent with the expression of pro-inflammatory cytokines. It is worth noting that the expression level of cytokines in the superinfection group significantly increased not only in the Bursa Fabricii but also in the thymus, and was more pronounced in the latter. This suggests that a superinfection with ALV-J and IBDV may induce a different pattern of immune injury than a single infection.

We assessed cellular immune responses by measuring the blood proportions of CD3^+^, CD4^+^, and CD8α^+^ cells. Previous studies have shown that ALV-J could induce lymphocyte apoptosis in immune organs, especially in young chickens [35]. Our results showed both ALV-J and IBDV infections induce the decline in the blood proportions of CD3^+^, CD4^+^, and CD8α^+^ at 21 dpi, and IBDV induced this effect more significantly. These results indicate that IBDV induces more severe lymphocyte apoptosis in peripheral blood. In the superinfection group, the proportions of lymphocytes exhibited a more severe decrease, although it was not more significant than the IBDV single virus-infected group. However, the ratio of CD4^+^/CD8^+^ was similar in the superinfection group when compared to the control. These results indicate that the superinfection of ALV-J and IBDV induces overall CD3^+^-positive lymphocyte apoptosis but does not change the ratio of CD4^+^/CD8^+^.

Our research demonstrates the effectiveness of a candidate model for studying the outcome of a superinfection with ALV-J and IBDV in SPF chickens. Using the infection model, we elucidated synergistic pathogenic effects between ALV-J and IBDV. In addition, we found that the immunosuppression induced by ALV-J infection combined with the immunosuppression and pathogenicity induced by IBDV infection can seriously impair the host’s immune resistance to both viruses’ infections, inevitably resulting in a more serious illness in the superinfected chickens. However, further studies are needed to elucidate the deeper mechanisms by which a superinfection of ALV-J and IBDV cooperates to induce more severe immunosuppression.

## Figures and Tables

**Figure 1 viruses-14-02312-f001:**
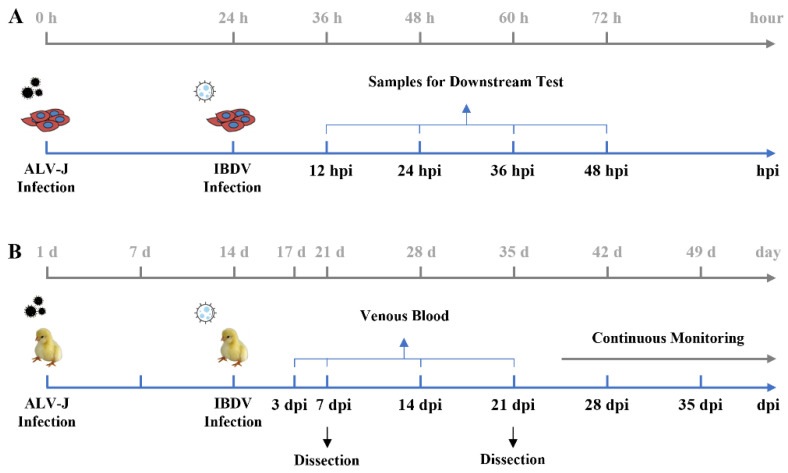
Time course diagram. (**A**) Time course of superinfection of ALV-J and IBDV in DF-1 and CEF cells. The cultured monolayer cells were infected with ALV-J firstly and IBDV 24 h later. After 12, 24, 36, and 48 h of IBDV infection, cell samples were collected for downstream testing. All the experiments were performed independently at least three times. Hpi—hours post-IBDV infection. (**B**) Time course of superinfection of ALV-J and IBDV in SPF chickens. The SPF chickens were infected with ALV-J firstly and IBDV 14 days later. After 3, 7, 14, and 21 days of IBDV infection, venous blood samples of 6 chickens in each group were collected to detect viral load. At 7 and 21 days of IBDV infection, samples of the immune organs including Bursa Fabricii and spleen were excised and weighed for 6 chickens in each group. Weight loss and mortality were continuously monitored throughout the experimental period. Dpi—days post-IBDV infection.

**Figure 2 viruses-14-02312-f002:**
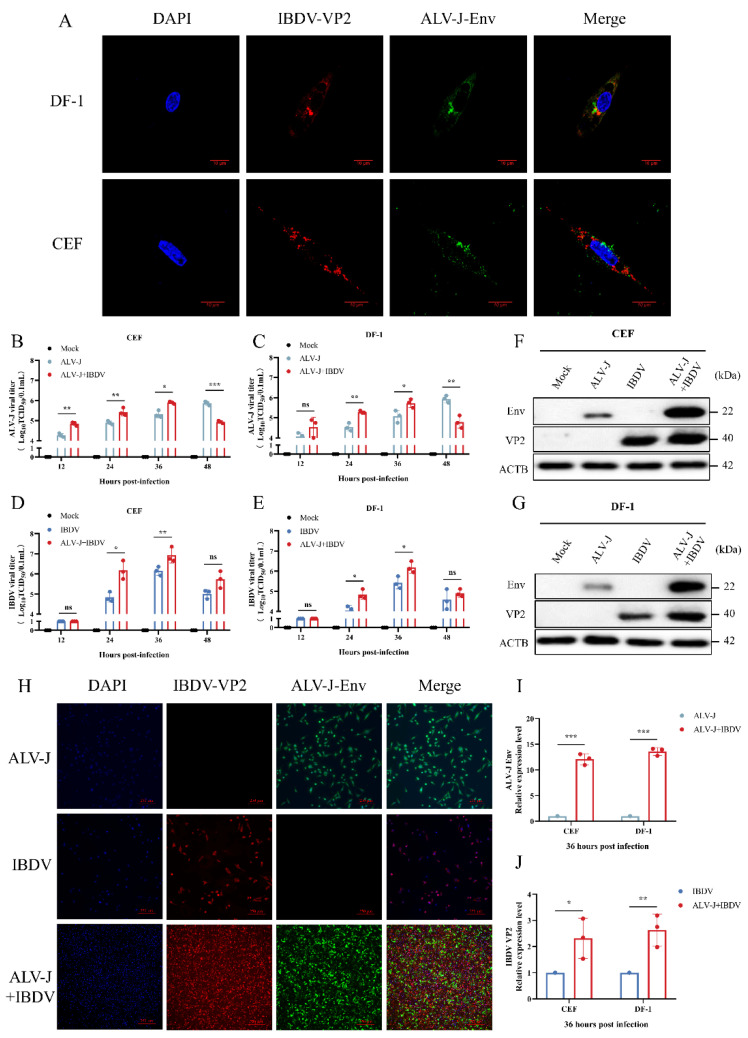
ALV-J and IBDV synergistically increase viral replication in vitro. (**A**) Protein expression and localization of ALV-J and IBDV examined in DF-1 and CEF cells by confocal laser microscope. Alexa fluor 488 (green for ALV-J Env) and Alexa fluor 594 (red for IBDV VP2) were used as the secondary antibodies in the assay. DAPI (blue) was used to stain the nuclear DNA. Scale bar: 10 μm. (**B**,**C**) ALV-J virus titers in CEF and DF-1 cells were determined from 12 hpi to 48 hpi by Reed–Muench methods, respectively. Data were expressed as mean ± SE and analyzed by One-way ANOVA test. (**D**,**E**) IBDV virus titers in CEF and DF-1 cells were determined from 12 hpi to 48 hpi by Reed–Muench methods, respectively. Data were expressed as mean ± SE and analyzed by One-way ANOVA test. (**F**,**G**) Protein expression levels of ALV-J env and IBDV VP2 in CEF and DF-1 cells, respectively, were examined by Western Blot at 36 hpi. (**H**) The protein expression of ALV-J env and IBDV VP2 were detected by immunofluorescence (magnification, ×10). Scale bar: 250 μm. (**I**) Densitometric ALV-J/ACTB ratios in CEF and DF-1 cells at 36 hpi are shown. (**J**) Densitometric IBDV/ACTB ratios in CEF and DF-1 cells at 36 hpi are shown. Data are expressed as mean ± SE and analyzed by Student’s *t*-test. These experiments were performed independently at least three times with similar results. *, *p* < 0.05; **, *p* < 0.01; ***, *p* < 0.001; ns, no significant.

**Figure 3 viruses-14-02312-f003:**
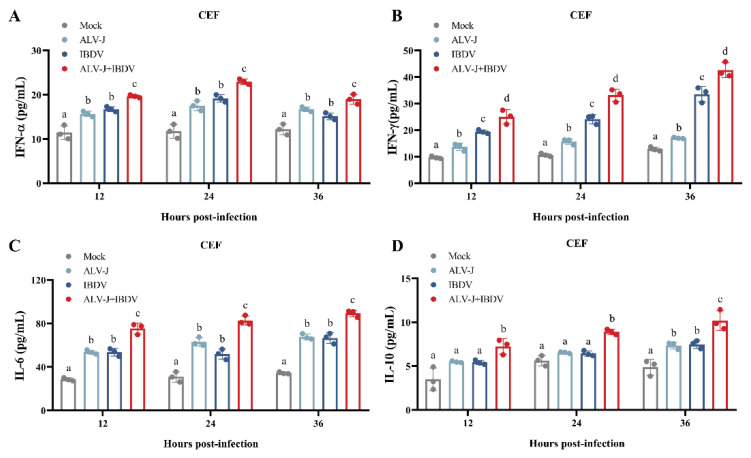
ALV-J and IBDV synergistically induce inflammatory mediator secretion in vitro. The secretion activities of inflammatory mediators IFN-α (**A**), IFN-γ (**B**), IL-6 (**C**), and IL-10 (**D**) were determined by ELISA in CEF cells from 12 hpi to 36 hpi. These experiments were performed independently at least three times with similar results. Data were expressed as mean ± SE and analyzed by One-way ANOVA test. Different lowercase letters indicate significant differences between different groups.

**Figure 4 viruses-14-02312-f004:**
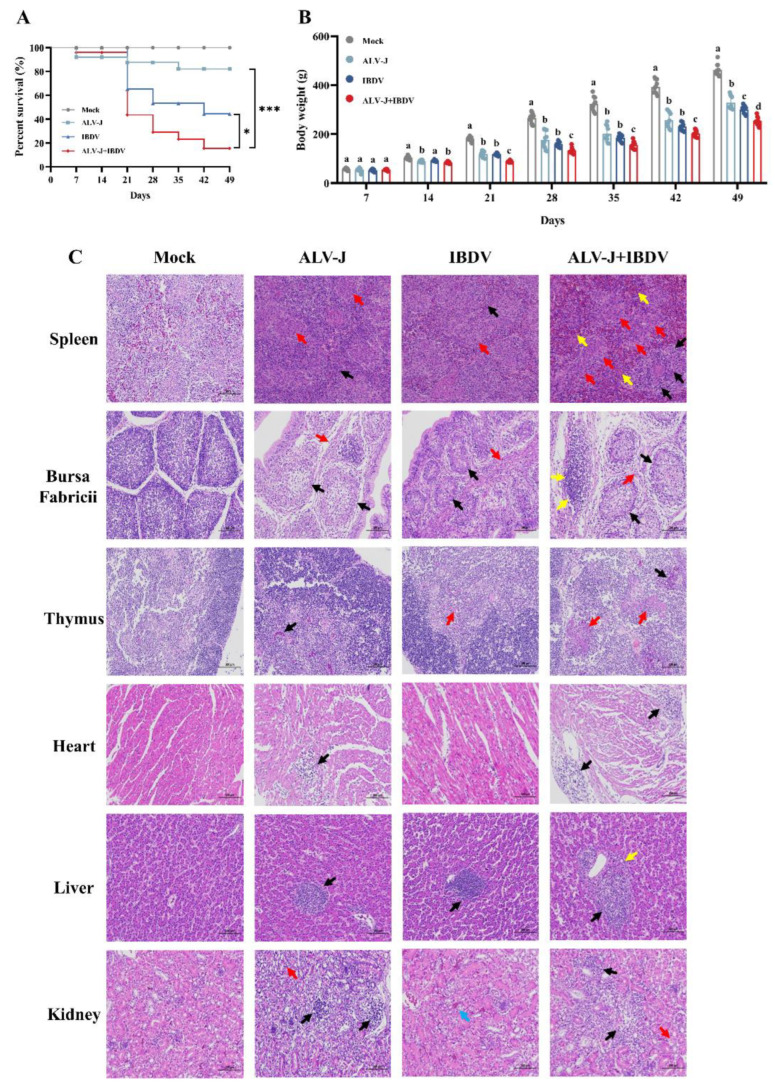
ALV-J and IBDV synergistically enhance pathogenicity in SPF chickens. (**A**) Survival curves for the Mock, ALV-J, IBDV, and ALV-J+IBDV groups. (**B**) Body weights of SPF chickens for each group from day 1 to day 49. Data are expressed as mean ± SE and analyzed by one-way ANOVA test. Different lowercase letters indicate significant differences between different groups. (**C**) Histological lesions of each group (hematoxylin-eosin, 200×). Spleen: lymphocytic depletion (black arrow), malignant reticulosis (red arrow), and polycythemia (yellow arrow). Bursa Fabricii: proliferation of undifferentiated epithelial cells (black arrow), connective tissue proliferation (red arrow), and inflammatory cell infiltration (yellow arrow). Thymus: vascular congestion (black arrow) and eosinophilic protein-like fluid exudates (red arrow). Heart: inflammatory cell infiltration (black arrow). Liver: inflammatory cell infiltration (black arrow) and heterophile leukocyte infiltration (yellow arrow). Kidney: inflammatory cell infiltration (black arrow), cytoplasmic loose (red arrow), and mild vascular congestion (blue arrow). The visual examinations of the histological lesions were controlled in a double-blind manner. *, *p* < 0.05; ***, *p* < 0.001.

**Figure 5 viruses-14-02312-f005:**
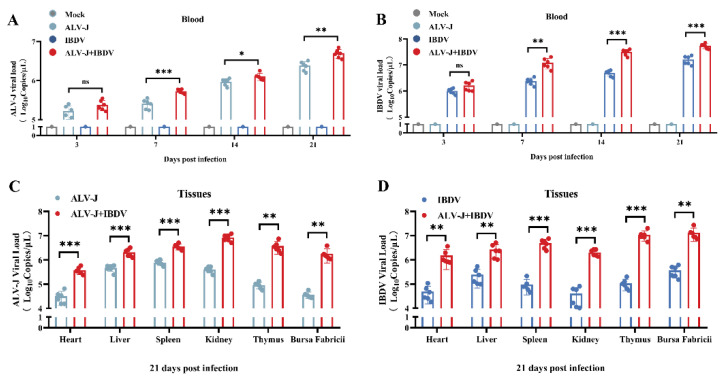
ALV-J and IBDV synergistically increase viral replication In Vivo. (**A**) ALV-J loads in blood from 3 dpi to 21 dpi. (**B**) IBDV loads in blood from 3 dpi to 21 dpi. (**C**) ALV-J loads in spleen, bursa, thymus, heart, liver, and kidneys at 21 dpi. (**D**) IBDV loads in spleen, bursa, thymus, heart, liver, and kidneys at 21 dpi. Data were expressed as mean ± SE and analyzed by One-way ANOVA test. *, *p* < 0.05; **, *p* < 0.01; ***, *p* < 0.001; ns, no significant.

**Figure 6 viruses-14-02312-f006:**
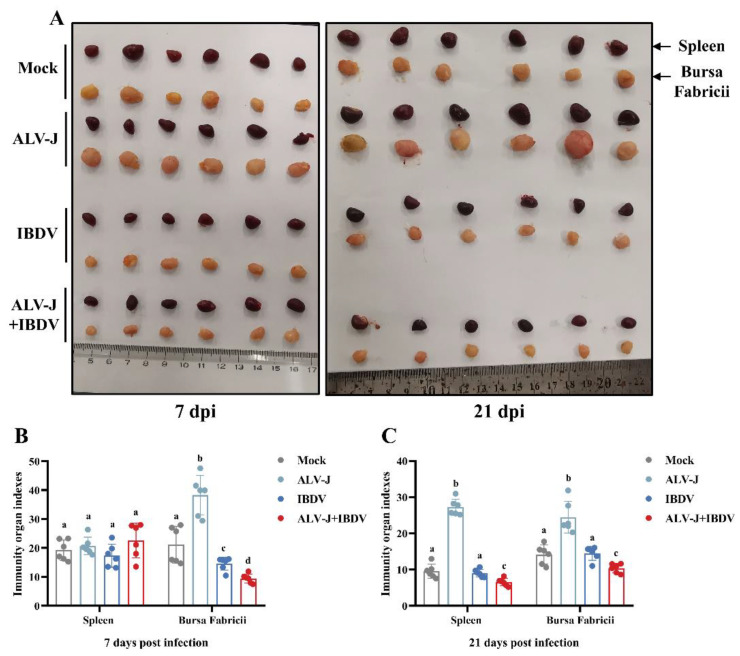
ALV-J and IBDV synergistically induce immune organ atrophy in SPF chickens. (**A**) Anatomical picture of spleen and bursa for the Mock, ALV-J, IBDV, and ALV-J+IBDV groups at 7 dpi (left) and 21 dpi (right). The top row is the spleen, and the bottom is the bursa fabricii. Six replicates of each group are shown. (**B**) Immune organ indexes of spleen and bursa at 7 dpi. (**C**) Immune organ indexes of spleen and bursa at 21 dpi. The immune organ indexes are expressed as the weight of immune organ relative to body weight. Data are expressed as mean ± SE and analyzed by One-way ANOVA test. Different lowercase letters indicate significant differences between different groups.

**Figure 7 viruses-14-02312-f007:**
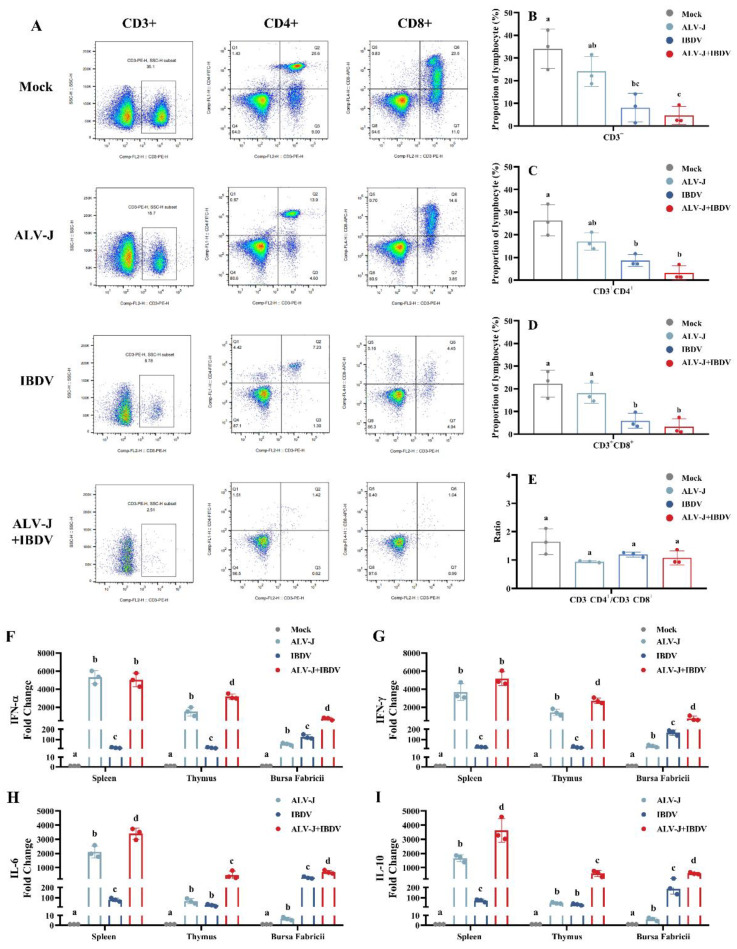
ALV-J and IBDV synergistically induce lymphocyte apoptosis and inflammatory mediator secretion in vivo. (**A**) Representative FACS scatter diagrams showing the percentages of CD3^+^, CD4^+^, and CD8^+^ cells in peripheral blood mono-nuclear cells from treatment groups ALV-J, IBDV, ALV-J+IBDV, and empty control. The proportion of lymphocyte sub-groups CD3^+^ (**B**), CD3^+^CD4^+^ (**C**), CD3^+^CD8^+^ (**D**), and the ratio of CD3^+^CD4^+^/CD3^+^CD8^+^ (**E**) were summarized in the diagram on the right. The secretions of inflammatory mediators IFN-α (**F**), IFN-γ (**G**), IL-6 (**H**), and IL-10 (**I**) were determined by RT-qPCR for the spleen, thymus, and bursa at 21 dpi. Data were expressed as mean ± SE and analyzed by One-way ANOVA test. Different lowercase letters indicate significant differences between different groups.

**Table 1 viruses-14-02312-t001:** Primer used for RT-qPCR.

Primer	Sequence (5′-3′)
Env-F	TGCGTGCGTGGTTATTATTTC
Env-R	AATGGTGAGGTCGCTGACTGT
VP2-F	ATGACAAACCTGCAAGATCA
VP2-R	ATCGAACTTGTAGTTCCCAT
IL-6-F	AATCCCTCCTCGCCTTTCTG
IL-6-R	GCCCTCACGGTCTTCTCCAT
IL-10-F	GCTCTGAGCACAGTCGTTTG
IL-10-R	CAGATGGGGACGTGGTTACG
IFN-α-F	CAACGACACCATCCTGGACA
IFN-α-R	ATCCGGTTGAGGAGGCTTTG
IFN-γ-F	GAGCCAGATTGTTTCGATGTACTTG
IFN-γ-R	CATCAGGAAGGTTGTTTTTCAGAG
GAPDH-F	GAACATCATCCCAGCGTCCA
GAPDH-R	CGGCAGGTCAGGTCAACAAC

## Data Availability

The data presented in this study are available on request from the corresponding author.

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
