# Peer review of "Synergistic Immunosuppression of Avian Leukosis Virus Subgroup J and Infectious Bursal Disease Virus Is Responsible for Enhanced Pathogenicity"

_viruses, 2022, doi:10.3390/v14102312_

Round 1
Reviewer 1 Report
This is an interesting paper that attempts to quantify responses of avian cells and chickens to ALV-J and IBDV, either singly or in combination. It is important research because of the implications for agriculture and the poultry industry as well as for the pure scientific interest. The strong points of the paper include the importance for developing cell culture and animal models to test the effects of these two avian pathogens and the demonstration of certain effects that suggest dual infection can give different effects than single infections. Overall, the experiments are clearly exhibited. The major weakness of the paper surrounds the assertion of true synergy in infection. In almost all of the experiments it is difficult to unequivocally assert that there is evidence of synergy, since it is difficult to determine whether or not cells have been coinfected. On the other hand, it may be possible that cell-to-cell paracrine communications among cells differentially infected could contribute to the implied synergy, and this scenario should be more thoroughly investigated/discussed.
A few specific points:
There are no figure numbers or legends in the manuscript, these must be added.
It would be worthwhile to define and describe the cell types used (CEF and DF-1) and explain why these cell lines were chosen for the experiments.
In description of the Histopathology assays, a description of what specific attributes of pathology are examined and whether the visual examinations for CPEs are controlled in a double-blind manner to avoid experimental biases based on expectations.
In Section 3.2., the statement and citations “Several important inflammatory mediators, such as IL-6, IL-10, IFN-α and IFN-γ, have been shown to be involved in immunosuppression and tumorigenesis19, 20.” require some clarification. Since these immune factors are actually largely involved in the instigation of immune response and several can be used as antitumor agents, the sentence does not really make sense. Furthermore, the references 19 and 20 are not terribly relevant to the very general statement that they are supposed to support.
The results of cytokine secretion in Figure 3 are interesting, but it is unclear whether the results are synergistic or just additive. The authors do show that it is possible for cells to be coinfected with both viruses, but they do not show that all cells are infected with both viruses. So it is possible that the increase in cytokine secretion into the media when both viruses are added to the cells is because more total cells get infected under these conditions. It is also worth noting that the some of these cytokines (mostly the interferons) would reasonably be expected to prevent or at least mitigate further spreading infection in cell culture. It looks like the dually infected cells are actually responding more strongly than the singly activated cells. To help more fully examine whether there is synergistic activity, it could be useful to transfer media from singly or doubly infected cells and use this to (pre-) treat fresh cells prior to single or double infection. Presumably the double infection (if synergistic and more potentially pathogenic) would cause cells to produce cytokines into the media that was less protective toward subsequent infection (when normalized for total cytokine levels secreted).
In the histopathology studies shown in Figure 4, the evidence of synergy is not very compelling, since there appear to be roughly comparable numbers of notable CPEs in the singly- versus doubly- infected mice. It would be more compelling if the authors were able to show with some immune staining that the CPEs were associated with antigens of each/both viruses.
Figure 5 is probably the most convincing experiment shown, as there does appear to be a significant difference in viral load emanating from singly vs. doubly infected cells.
The flow cytometry shown in Figure 7 seems to suggest that the effects of infection by ALV-J are masked by those of IBDV, i.e., it is difficult to surmise any cooperative or synergistic effect, just a dominant effect by IBDV.
Author Response
Thanks for your time to review our manuscript. We have carefully considered the comments and revised the manuscript accordingly. The following is a point-by-point response to all those comments and a list of changes we have made in the revised manuscript.

Reviewer 2 Report
the manuscript investigates the possible synergestic effect between ALV-J and IBDV viruses after double infection either in in-vitro (using DF-1 or CEF cells) or in-vivo in SPF chicken. The authors showed that double infection with both viruses has a synergestic effect on terms of both viruses replication, immune response, and growth rate.
as the authors mentioned in the introduction, double infection is a common event in real (chicken) life. the pssibilities that chicken can be infected with two viruses are high and the consequences are usually dramatic.
of course, it is always good to shed light on such events and runs some investigations from time to time.
my main comment is related to materials and methods: The MM section is not well-written. The authors missed important details all over this section. for example, section 2.2. does not convey the actual experimental design mentioned later on in the results. Groups of infection are not clearly described. It is not clear how the authors have calculated virus titers using syber-green qPCR. Statistical analysis is very brief; I do not think that Student's t-test is sufficient to run all statistical analysis for all trials giving that several groups need to be compared with each other's. overall the MM section, more details need to be provided (what samples were collected at what time, more details for immunofluorescence and confocal microscopy, histopathological section needs proper re-writing).
It is better when each fgure legend comes after its figure. figure legends will need more details on what is in the figure; for example: which microscope was used, lens magnifications, which statisitcal analysis was used and p-values.
Author Response

(The authors gave the same response as above.)

Reviewer 3 Report
The authors have conducted in vitro and in vivo studies with a view of establishing a superinfection model and study the immunosuppression in chickens. The study has been done in a systematic manner. However, data analyses have been poor as has been the writing the manuscript.
1) What will be the outcome if IBDV infection is done first?
2) In vitro exp: One important piece of information is missing what is the cell viability following infections? The cell viability data for all the time points should be given.
3) How did you come with MOIs used in the in vitro exps?
4) In vitro study: How many replicates per group and how many times the study was repeated?
5) There are multiple groups and multiple time points, T test is a wrong test to use (not acceptable). Therefore, all the stat analyses should be done after consulting a biostatistician.
6) There are no figure legends as such, it is very difficult to understand what the figures, and illustrations mean
7) All the reagents used should be given the company name followed by city, state/province and country.
8) Also, all the antibodies used: Are they anti-chicken?
9) Antibody concentrations and dilutions should be given and not the volumes.
10) Histology: Need to quantify the lesions and do stat analyses.
11) Histology figures: Define what the arrows are indicating?
12) Immune organs: what the columns mean?
13) Author contributions: Author name initials should be given rather than whole names.
14) Data availability statement is wrong.
15) I have not seen authors acknowledge reviewers.
Author Response

(The authors gave the same response as above.)

Round 2
Reviewer 1 Report
I feel the authors did a very thorough job of revising their manuscript and paid careful attention to reviewers' comments.
Author Response
Thanks for your time again!
Reviewer 2 Report
thank you for the revised version. the manuscript improved significantly and my comments were addressed properly
Author Response
Thanks for your time again!
Reviewer 3 Report
Still some of the comments are not addressed adequately.
1) Figure 2B-E and Figure 5 A and B: There are >2 groups at each time point and t test can not be used and not acceptable
2) If the authors go to Southern Biotech (https://www.southernbiotech.com/), all the chicken antibodies are available for last two decades. The antibodies non specific to chicken markers may be working but it should be discussed in the discussion section.
3) Did authors use iso type controls for each of the markers? What were the iso type control?
4) How long you incubated the cells with antibodies? The, what did you do?
Author Response
Thanks for your time again.
